# Evaluating the sustainability of differentiated service delivery interventions for stable ART clients in sub-Saharan Africa: a systematic review protocol

Nwanneka Ebelechukwu Okere [ORCID],[1] Lisa Urlings,[2] Denise Naniche,[3] Tobias F Rinke de Wit,[1] Gabriela B Gomez,[4] Sabine Hermans[1]

GBG and SH are joint senior authors.

For numbered affiliations see end of article.

**Correspondence to**
Ms Nwanneka Ebelechukwu Okere; n.okere@aighd.org

## ABSTRACT

**Introduction** In 2015, WHO recommended immediate treatment for people living with HIV (PLHIV). As a result, the number of PLHIV needing antiretroviral therapy (ART) in sub-Saharan Africa (SSA) doubled from 12 million to over 25 million. This put a strain on already weak health systems and inspired the exploration of innovative service delivery models—differentiated service delivery (DSD). In DSD, services are tailored according to client clinical type and offer much-needed improvement in efficiency. The potential of achieving good outcomes for both clients and the health system plus the promise of sustainability motivates DSD promotion especially in low-income and middle-income countries. This review aims to evaluate the sustainability of DSD interventions.

**Methods and analysis** We will systematically review peer-reviewed English literature published between 2000 and 2019 identified by searching PubMed and EMBASE databases. Main inclusion criteria comprise studies describing DSD interventions conducted in SSA focused on stable adult ART clients, whether described alone or compared with clinic-based service delivery. Quality of included studies will be assessed employing the Down and Black's and Joanne Briggs Institute checklists for quantitative and qualitative studies, respectively. We will apply a comprehensive sustainability framework including 40 individual constructs to evaluate, score and rank each intervention for sustainability. Narrative and quantitative synthesis will be conducted as appropriate.

**Ethics and dissemination** No ethical approval is required for this study as it is a review of published or publicly available data. Review results will be published in a peer-reviewed journal and presented at international conferences.

**PROSPERO registration number** CRD42019120891.

### Strengths and limitations of this study

► The current systematic review will assess retrospectively the sustainability of differentiated service delivery (DSD) models for HIV treatment in terms of their impact.

► We aim to equip policy-makers with information necessary for making rational choices to prevent wastage of limited resources by providing an evaluation of the sustainability of DSD interventions.

► The comparability of results could be a challenge because a wide range of interventions have been reported and consequently a wide range of outcomes are expected to be included.

► We will review interventions conducted in sub-Saharan Africa and articles published in English leading to a possible bias by excluding articles in other languages.

(SSA) more than doubled from 12 million to over 25 million.[1][2] This increase in demand adds strain on already weak health systems. It made obvious the insufficiency of traditional clinic-based care and called for innovative service delivery models. Differentiated service delivery (DSD) prioritises client preferences while also aiming to increase efficiency.[3–8] DSD has evolved over time and encompasses a wide range of concepts in programme implementation especially in low-income and middle-income countries such as task shifting, decentralisation and community-based ART. Ultimately, the goal of DSD is to increase access to and retention in care by reducing demand-side and supply-side barriers of clinic-based models of service delivery, especially for rural populations.[9–12]

Differentiated care became necessary as more clients were initiated on lifesaving antiretrovirals and ART clinics were becoming

## INTRODUCTION

In 2015, WHO recommended treatment initiation for people living with HIV (PLHIV), a 'universal test and treat' strategy. As a result, the number of PLHIV needing antiretroviral therapy (ART) in sub-Saharan Africa

crowded. This largely influenced the public health approach involving decentralisation recommended by WHO as far back as 2006.[13 14] Since then, attempts have been made to streamline services by categorising clients in terms of their clinical profile, such as being 'stable'. Currently, WHO defines a stable client as: 'being on ART for at least 1 year, no current illnesses or pregnancy, a good understanding of lifelong adherence and evidence of treatment success (two consecutive viral load measurements below 1000 copies/mL)'.[1]

Four DSD models were recently described which focus on stable ART clients.[15 16] They include (1) healthcare worker (HCW) groups, (2) facility-based individual models, (3) client managed groups and (4) out-of-facility individual models. Each model is defined by four dimensions, namely 'who' provides care, 'what' care is provided, 'when' do the activities happen and 'where' is care provided. Prepackaged ART refill, symptom screening, health talks, clinical consultations or referral if unwell defines 'what' care is provided across models. 'When' care is provided, as a major benefit of DSD, is usually less frequent than the routine monthly visits for clients categorised as stable. Though WHO recommends 3–6 months intervals, anything between 2 and 12 months is possible depending on the context. But these models are characterised by the other two dimensions, that is, 'who' provides care (the cadre of healthcare personnel involved) and 'where' care is provided (location). Interventions may be conducted by HCWs within facilities, for example, adherence clubs (ACs), fast-track ART refill, 6-monthly appointments, multimonth scripting or by community-based HCWs, for example, community drug distribution points. Interventions can also be conducted by non-HCWs within facilities, for example, ACs or by community-based non-HCWs, for example, community ART refill groups, community ART groups, mobile outreach (OR) and AC (see table 1).[17]

The potential for achieving a lasting impact and sustainability is the push behind DSD promotion. Sustainability in healthcare is a broad concept which was only recently defined as 'programme, clinical intervention and/or implementation strategies which continue to be delivered after a defined period of time (ie, after expiration of initial external funding) including the maintenance of programme or individual behavioural change (ie, clinician, client) which may evolve or adapt while continuing to produce benefits for individuals/systems'.[18] Several conceptual frameworks have been described articulating the constructs mentioned in this definition.[19–25] These frameworks consolidate and elaborate on sustainability as an important concept embedded within the continuum of implementation science.[26–28] Most of them view sustainability as a process to be prospectively explored during implementation rather than an outcome to be retrospectively evaluated. Lennox et al,[29] in a recent review, developed a comprehensive framework of 40 individual sustainability constructs categorised into six broad domains, namely (1) intervention design and delivery, (2) intervention processes, (3) external environment, (4) organisational setting, (5) resources and (6) people involved. The Lennox framework is the basis used in this review.

## Why is it important to do this review?

The long-term sustainability of HIV programme interventions is essential as lifelong ART remains a necessity for PLHIV. Therefore, sustainability underlies strategies endorsed by HIV programme, for example, task shifting, decentralisation of care, differentiation of care.[15 30–35] The increasing international call for universal health coverage is similarly motivated (in addition to other considerations such as equity) and will likely benefit not just HIV programme but entire health systems.[36 37] While many DSD interventions have been implemented within HIV programme with good outcomes, knowledge about their sustainability has been limited.[38–40] Furthermore, it is not clear how these interventions can be properly integrated within the existing health system to ensure they last.

Research on the sustainability of health interventions has been hampered by a lack of consensus. For example, several definitions have existed, some referring to the continuation of an intervention as a whole or in part[41]

| Table 1 | Characterisation of DSD models | | | |
|---------|------|------|------|------|
| | | Location (where) | | |
| | | Within facilities | Community based | |
| Personnel (who) | HCW managed | Fast-track ART refill Six-monthly appointment Multimonth scripting Adherence clubs (ACs) (model 1) | ACs Community drug distribution points (model 2) | |
| | Non-HCW managed | ACs (model 3) | Community ART groups Community ART refill groups ACs Mobile outreach (OR) (model 4) | |

ART, antiretroviral therapy; DSD, differentiated service delivery; HCW, healthcare worker.

and others to the continuation of benefits of an intervention.[42] Similarly, several sustainability frameworks exist, making comparability across studies difficult. Common grounds are, however, beginning to emerge. The framework by Shediac-Rizkallah and Bone[43] provided the basic constructs on which other sustainability studies continue to build.[44–46] The recent syntheses of a comprehensive definition and framework, both anchored in this framework, provided useful tools which we have adopted to gain insight into the sustainability of DSD models.[18 29]

In resource-constrained settings, the promising client-related outcomes and the reduced demand on resources it places on health systems without undermining effectiveness are thought to make DSD feasible to implement. HIV programme will, therefore, require evidence to inform which DSD options to explore. Our review aims to fill this gap by evaluating the sustainability of available DSD interventions.

While DSD interventions may not be as onerous as standard clinical care, they pose some demands to the health system. As countries in SSA strive to increase their domestic share of funding for their HIV programmes, they must be aware of the aspects influencing sustainability of the DSD options considered. This review will provide information necessary to inform these discussions. Our review also aims to provide the impetus for further enquiry into service delivery models and socioeconomic issues necessary to ensure continual client empowerment while supporting minimally resourced health systems.

## METHODS AND ANALYSIS

The protocol adheres to the Preferred Reporting Items for Systematic Reviews and Meta-Analyses Protocols guidelines.[47]

### Search strategy

We will search PubMed and EMBASE to identify eligible articles using a comprehensive search strategy (see online supplementary file I). Additionally, the reference lists of included articles and the reference lists of previously published reviews will be reviewed for other relevant articles. The review period of interest will be between 2000 and 2019 to capture all articles published since the widespread scale-up of ART services across SSA. Only English language articles will be included. We will use Rayyan Qatar Computing Research Institute (Rayyan QCRI) app, the free systematic reviews web application[48] to store, organise and manage eligible references whose titles and abstracts will be screened. A follow-up search will be conducted prior to the completion of the review which is planned 9 months after commencement.

### Eligibility criteria

The inclusion and exclusion criteria of articles are as listed— see box 1. Appropriate definitions to guide the selection of articles are also provided—see box 2.

---

**Box 1  Eligibility criteria**

**Inclusions**
► Observational, experimental or quasi-experimental studies.
► Studies involving stable adult antiretroviral therapy clients accessing HIV care in sub-Saharan Africa.
► Studies describing or assessing HIV services delivered through models other than standard clinic-based care.
► Studies which compare the performance of these other service delivery models with standard clinic-based HIV service delivery accessed by other clients. Though, lack of this comparison is not an exclusion criterion.

**Exclusions**
► Reviews, editorials, protocol studies and clinical guidelines.
► Studies describing or assessing interventions focussed on special population groups for example, adolescents, children, pregnant women, men who have sex with men, commercial sex workers.
► Studies utilising data retrospectively collected in electronic databases with little description of the actual intervention.

---

### Selection of studies

Two reviewers (NEO and LU) will independently screen the titles and abstracts of articles identified in the searches to identify potential articles. The reviewers will then conduct full-text reading of these potential articles to determine eligibility. GBG and SH will be engaged in discussions to make a final determination on eligibility whenever the two reviewers are unable to reach a consensus. Reasons for exclusions will be reported.

### Data extraction

We have developed a data extraction form. NEO and LU will pilot the data extraction form on a subset of five articles to assess its functionality. After incorporating feedback obtained from the review team (NEO, LU, GBG and SH), a final form will then be used to extract relevant data independently by the two reviewers.

Data to be extracted will include: (1) details of each intervention including site, setting, service delivery model, funding, providers, period, theoretical underpinning; (2) study population, participant type and numbers; (3) sustainability constructs as described in table 2 and (4) challenges or unintended consequences faced during implementation, comments, clients and staff perspectives (see online supplementary file II).

---

**Box 2  Definitions**

**Stable antiretroviral therapy (ART) clients**
Definition of a stable ART client has been fine-tuned over time, therefore, there will be no restriction in terms of definition. We will rely on the studies' definitions and will provide a summary of definitions employed.

**Adult clients**
The definition of adult clients varies across studies. We will define adult as ≥18 years but will include studies if the age range of participants is not < 15 years.

---

**Table 2** Sustainability constructs and evidence to be extracted (adapted from Lennox et al[29]) (primary study outcomes)

| Domain | Item | Sustainability construct | Outcome | Evaluation question |
|---|---|---|---|---|
| The intervention design and delivery | 1 | Demonstrating effectiveness | Patient-related outcomes | Does the paper report any numeric or subjective patient-centred outcomes to show effectiveness, for example, retention-in-care, viral suppression, lost to follow-up, patient satisfaction? |
| | 2 | Evidence base for the intervention | Evidence base | Is there evidence that the intervention provides the expected benefits as planned that is, that the DSD improves outcomes? |
| | 3 | Expertise | Expertise | Is there evidence of adequate expert knowledge and experience to carry out the DSD especially by supporting organisation? |
| | 4 | Quality improvement (QI) methods | QI methods | Is there evidence that QI methods that is, using data to identify gaps which are continually improved, starting with a pilot and then spreading. Are used to support intervention success and sustainability? |
| | 5 | Monitoring progress | Monitoring progress | Is there a standardised and systematic method to gather and report data during DSD intervention? |
| | 6 | Intervention duration | Duration | Is there evidence that the intervention will last beyond initial funding? |
| | 7 | Intervention type | Project design | What type of intervention is it, for example, prevention, treatment, palliative, supportive care? |
| | 8 | The problem | Problem awareness | Is there general awareness of a problem among stakeholders that requires the DSD intervention to address? |
| | 9 | Training and capacity building | Capacity building | Is there evidence of any orientation, training, ongoing mentoring for staff delivering the DSD intervention? |
| The external environment | 10 | Awareness and raising the profile | Community awareness | Is there evidence of the larger community being aware of the DSD intervention and promoting its benefit? |
| | 11 | Socioeconomic and political considerations | Political support | Is there evidence that the intervention has political support? For example, government engagement, guidelines revision to include DSD requirement? |
| | 12 | Spread to other organisations | Spread | Is there evidence that the intervention or underlying concepts spread within participating organisation or to other locations? |
| | 13 | Urgency | Urgency | Is there evidence of an urgency to maintain the intervention based on its relevance? |
| Intervention processes | 14 | Accountability of roles and responsibilities | Roles and responsibilities | Is there evidence that roles and responsibilities of staff involved in the DSD are spread out and clearly defined? |
| | 15 | Belief in the intervention | Belief in intervention | Is there evidence that staff think the DSD intervention is a better way to do things? |
| | 16 | Complexity | Complexity | Is there evidence that it is not difficult for staff to understand and conduct the intervention? |
| | 17 | Defining aims and shared vision | Shared goal | Is there evidence of a shared aim and vision established with all stakeholders before commencing the intervention? |
| | 18 | Incentives | Motivation | Is there evidence that rewards, or benefits derived from the DSD intervention are considered enough motivation that drive stakeholders to engage and continue delivering intervention over time? |
| | 19 | Job requirements | Job requirements | Is there evidence of revision of job requirement for key staff incorporating the DSD intervention tasks as part of key job descriptions? |
| | 20 | Workload | Workload | Is there evidence that any additional workload introduced by the DSD intervention is manageable and requiring no special effort to staff involved? |
| Resources | 21 | General resources | General resources | Is there evidence that resources needed to manage and maintain the DSD intervention are available? |
| | 22 | Funding | Funding | Is there evidence that adequate funds are available to implement, and strategic funds planned to sustain intervention that is, DSD will be embedded and sustained? |
| | 23 | Infrastructure | Infrastructure | Is there evidence that resources required to support the DSD intervention, for example, office space, materials and supplies are available? |
| | 24 | Staff | Staff | Is there evidence of enough staff in place to conduct and sustain DSD intervention? |
| | 25 | Time | Time | Is there evidence that adequate time was dedicated for DSD intervention in the routine daily schedule of the facility? |

Continued

| Domain | Item | Sustainability construct | Outcome | Evaluation question |
|--------|------|--------------------------|---------|---------------------|
| Organisational setting | 26 | Integration with existing programmes and policies | Integration | Is there evidence that DSD intervention was embedded within the existing organisational structure, programmes and policies? |
| | 27 | Intervention adaptation and receptivity | Adaptation | Is there evidence that the DSD intervention is flexible to respond, change, adapt and fit with local context requirement? |
| | 28 | Opposition | Opposition | Is there evidence of any resistance due to other competing interests from stakeholders reported? |
| | 29 | Organisational readiness and capacity | Readiness | Is there evidence that health facilities have adequate capacity and readiness to undertake the DSD intervention that is, in terms of materials and manpower? |
| | 30 | Organisational values and culture | Values and culture | Is there evidence that the values of the DSD intervention align with health system values, prevailing beliefs and culture and priorities? |
| | 31 | Support available | Management support | Is there evidence of facility management support for the delivery and maintenance of the DSD intervention? |
| The people involved | 32 | Leadership and champions | Champions | Is there evidence of any influential person or group who advocates and supports the DSD intervention? |
| | 33 | Ownership | Ownership | Is there evidence that stakeholders take ownership to support, embed and sustain the DSD intervention? |
| | 34 | Power | Power | Is there evidence that stakeholders can use their power to make decisions, advocate and support the DSD intervention? |
| | 35 | Relationships, collaboration, networks | Collaboration | Is there evidence of any collaborations, partnerships and support networks to promote and sustain the DSD intervention? |
| | 36 | Satisfaction | Satisfaction | Is there evidence of benefits and rewards enjoyed by stakeholders and staff for participation in DSD intervention reported? |
| | 37 | Stakeholder participation | Stakeholder participation | Is there evidence that key stakeholders (those affected by the intervention) are engaged and participate in DSD intervention? |
| | 38 | Community participation | Community participation | Is there evidence of the participation of community members in directing and shaping DSD intervention goals and approaches to reflect their values and needs? |
| | 39 | Patient involvement | Patient involvement | Is there evidence of the involvement of patients in DSD intervention processes to understand patient's perspectives, values and needs? |
| | 40 | Staff involvement | Staff involvement | Is there evidence of the involvement of staff in the planning, design, delivery of the DSD intervention? |

DSD, differentiated service delivery.

We speculate that information regarding some constructs in the framework may be under-reported in the included articles. However, by describing the extent to which this under-reporting is happening, we hope to highlight this issue for further research. We will quantify and report the levels of missing information observed during data extraction (estimated by the number of 'not described' (ND) assigned by reviewers) per construct for included studies.

### Data analysis

Primary outcome measures will include the scores and rankings on sustainability constructs and domains as defined by Lennox *et al*[29] and presented in table 2. Since the review aims to extract and appraise evidence of sustainability constructs retrospectively, we simplified a previously described tool to assess sustainability in ongoing projects.[49] The two reviewers will assign a score of 3, 2 or 1 independently based on their assessment of whether there was enough, some or no evidence about each construct (see online supplementary file II).

Additionally, "ND" will be assigned if the reviewers are not able to make an assessment with information provided. Attempts will be made to contact study authors directly for any additional data considered necessary. Secondary outcome measures will include (1) trends observed in sustainability scores as a result of variations in the criteria employed to define stable clients, (2) if appropriate, quantitative intervention outcome measures such as retention in care, viral suppression, lost to follow-up, client or provider costs reported across similar interventions and (3) qualitative outcomes such as perspectives of clients on issues such as satisfaction with intervention or HCW, access to ART, peer support and perspectives of HCWs on workload or care for sick clients among others. Qualitative outcomes will be extracted as text.

Qualitative and quantitative analyses will be conducted. The basic characteristics, key findings including strengths and challenges of DSD interventions will be summarised. For qualitative analysis, we will conduct narrative synthesis[50] to identify commonly occurred themes. For

quantitative analysis, we will employ descriptive statistics to determine the sustainability performance of DSD interventions. Performance will be measured in terms of overall construct scores and percentages and median domain scores. This will then be used to rank constructs, domains, interventions and models for sustainability with higher scores ranked as better sustainability.

### Quality of evidence

Two reviewers, NEO and LU, will assess risk of bias in included studies using the Downs and Black checklist.[51] The following domains will be assessed: quality of reporting, external validity and internal validity, confounding (selection bias) and power. For qualitative studies, we will assess risk of bias using the Joanna Briggs Institute checklist.[52] Disagreements between reviewers will be resolved as for study selection. The overall quality of included studies will be evaluated, if possible, using the Grading of Recommendations Assessment, Development and Evaluation framework for quality of evidence.[53]

## ETHICS AND DISSEMINATION

Results and findings of the review will be published in a peer-reviewed journal and presented at appropriate conferences and meetings.

**Author affiliations**
[1]Global Health, University of Amsterdam, Amsterdam Institute for Global Health and Development, Amsterdam, The Netherlands
[2]Medicine, Amsterdam University Medical Centres, Amsterdam, Noord-Holland, The Netherlands
[3]Global Health, University of Barcelona, Barcelona Institute for Global Health, Barcelona, Catalunya, Spain
[4]Global Health and Development, London School of Hygiene and Tropical Medicine, London, UK

**Contributors** NEO, GBG and SH conceived the review design and in refining and registering the review protocol. NEO, GBG, SH and LU contributed to the search strategy. NEO drafted the initial manuscript. NEO, GBG, SH, DN and TFRdW provided content expertise. All authors contributed to revisions to obtain the final manuscript. All authors read and approved the final manuscript.

**Funding** This work was supported by Erasmus Mundus Joint Doctorate Fellowship, Framework Partnership Agreement 2013-0039, Specific Grant Agreement 2014-0681.

**Competing interests** None declared.

**Patient and public involvement statement** We will be reviewing and assessing already conducted interventions retrospectively and this will permit little or no involvement of patients and the public.

**Patient consent for publication** Not required.

**Ethics approval** No ethical approval is required for the conduct of this systematic review since all data used are available in the public space.

**Provenance and peer review** Not commissioned; externally peer reviewed.

**ORCID iD**
Nwanneka Ebelechukwu Okere http://orcid.org/0000-0001-9182-6518

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
