## [Reviewer comments · BMJ Open]

ARTICLE DETAILS

TITLE (PROVISIONAL)	Evaluating the sustainability of differentiated service delivery interventions for stable ART clients in sub-Saharan Africa: A systematic review protocol
AUTHORS	Okere, Nwanneka; Urlings, Lisa; Naniche, Denise; Rinke de Wit, Tobias; Gomez, Gabriela; Hermans, Sabine

VERSION 1 – REVIEW

REVIEWER	Nadia A Sam-Agudu 1. Institute of Human Virology Nigeria; Nigeria 2. Institute of Human Virology, University of Maryland School of Medicine, USA
REVIEW RETURNED	24-Oct-2019

GENERAL COMMENTS	The interrogation of the sustainability of DSD strategies in HIV programs as the topic of this systematic review is needed. It is also quite timely, given the plethora of DSD strategies and their adaptations in different contexts. DSDs have been the focus of the impact evaluation aspects of a significant number of implementation research studies published in the past 10 years. Some of these strategies have been the subject of economic analyses to establish their cost-effectiveness, which ideally should take their sustainability into consideration, but that is not often the case. We often project what sustainability should/is expected to be, and so this systematic review's retrospective evaluation of sustainability will be quite informative. I have mostly minor comments for consideration. GENERAL: 1. Replace the word "patient" with "client" as much as possible2. Authors should pay attention to a few grammatical errors such as the use of plural "s" when using a word as an adjective and not a noun eg symptom not symptomS screening, 3-6 month intervals, not 3-6 monthS intervals, HIV program (not programS) implementation, and so on3. The reader gets the sense in the Abstract and Intro that DSDs are only community-based, or are defined by being based outside of facilities. Please slightly revise the narrative to shift that interpretation4. The goals of this paper are highly relevant to those of implementation science/research, however I see very little reference to that discipline of research that focuses on service implementation, fidelity and sustainability in health systems and public health. Please infuse your narratives with some referencing to IS/IR, especially where you also discuss conceptual frameworks, including those that examine sustainability-implementation science has produced quite a
--

	few of those. ABSTRACT: 1. Line 17-20: suggest-we will systematically review peer-reviewed English-language literature published between 2000 and 2019. Our searches and inclusion/exclusion criteria were developed to identify studies focusing on (or about) DSD in SSA that included stable *adult* ART clients. STRENGTHS AND LIMITATIONS 1. These should be clearly sub-divided or narrated to show what is the stated strength and what is the stated limitation. 2. May be helpful to state that your review focuses on the sustainability of evidence-based DSDs in terms of prior evidence available on their actual impact with regard to patient outcomes? INTRO 1. Line 5-6, suggest: In 2015, the WHO recommended treatment initiation for all people living with HIV (PLHIV), the so-called "universal test and treat" strategy. 2. Line 11, suggest: DSD prioritizes patient preferences while also *aiming* to increase health system efficiency 3. Line 36: "when" care is provided *as* a major benefit... 4. Page 6, Table 1 title: Characterization of DCM Models." DCM is an undefined acronym at this point, also it stands for Differentiated Care Models and so the word "models" after that is not needed. I suggest sticking to "DSD models". 5. Line 46-47: Suggest: "until a vaccine *and/or cure * is developed...." 6. Eligibility criteria: these could be itemized in a bullet point on a Box Table for easy reference. I suggest definition of adult should be provided by a specific age range. Also, even though no restriction on definition of stable patient, authors can state they will rely on what the reviewed studies define as stable patient and with the actual review, provide some summary of what studies defined as stable patients. 7. Page 13, line 5: as countries in SSA strive to increase *the/their domestic* share of funding *for* their HIV programs... Supplementary Files 1. Search strategy: I assume authors will expand the search period beyond September 2018, when they actually perform the search? Eg expand closer to Dec 31, 2019? REFERENCES 1. I only found one reference document published in 2018. All other refs published before that. I expect there are more papers and DSD literature/guidelines/reports published in 2018 and 2019 that are relevant to the protocol under review. Please identify these and add for more recent context.
--	---

REVIEWER	Elvin Geng (Office) WUSTL, USA
REVIEW RETURNED	27-Oct-2019

GENERAL COMMENTS	I think this is an important systematic review to undertake and the authors have a concrete existing framework for sustainability to draw from. The only issue I have with this protocol is that at the information that the authors are looking for is likely under reported in the manuscripts, and the most likely outcome of the review is that
---

	sustainability is unclear because determinants of sustainability are rarely described. That would be a valuable observation, but more about the state of research in public health than about DSD per se. In fact, the authors should consider taking a random sample of papers to review in random order and if in the first 15 or 20 papers the information sought is vastly under-reported, the project could be stopped or it could pivot to answer a slightly different question.
--	---

VERSION 1 – AUTHOR RESPONSE

A. Reviewer #1:

General

Comment 1. *Replace the word "patient" with "client" as much as possible*

We have replaced the word patient with client as appropriate, except for those sentences when referring to patient-related outcomes or patient involvement. **Comment 2.** *Authors should pay attention to a few grammatical errors such as the use of plural "s" when using a word as an adjective and not a noun eg symptom not symptomS screening, 3-6-month intervals, not 3-6 monthS intervals, HIV program (not programS) implementation, and so on*

We have proofread the document and made the appropriate grammatical changes, eliminating 's' when the word is used as an adjective. Changes are noted in the marked document (pages 3, 4, 5, 6, 12, and tables) **Comment 3.** *The reader gets the sense in the Abstract and Intro that DSDs are only community-based, or are defined by being based outside of facilities. Please slightly revise the narrative to shift that interpretation*

We have modified the introduction section and the abstract to clarify that we refer to DSD as heterogeneous, a service model that can be within facilities as well as community based. **Comment 4.** *The goals of this paper are highly relevant to those of implementation science/research, however I see very little reference to that discipline of research that focuses on service implementation, fidelity and sustainability in health systems and public health. Please infuse your narratives with some referencing to IS/IR, especially where you also discuss conceptual frameworks, including those that examine sustainability-implementation science has produced quite a few of those.*

- The feedback has been incorporated by including the sentence "These frameworks consolidate and elaborate on sustainability as an important concept embedded within the continuum of implementation science" with appropriate references in line 13-15 on page 5 of the introduction to elaborate sustainability as an important concept within implementation science.

Abstract

Comment 1. *Line 17-20: suggest-we will systematically review peer-reviewed English-language literature published between 2000 and 2019. Our searches and inclusion/exclusion criteria were developed to identify studies focusing on (or about) DSD in SSA that included stable *adult* ART clients.*

- Sentence suggestion incorporated with slight modification in the second sentence for clarity in lines 13-18 i.e. "We will systematically review peer-reviewed English literature published between 2000 and 2019 identified by searching PubMed and EMBASE databases. Main inclusion criteria comprise, studies describing DSD interventions conducted in SSA focused on stable adult ART clients and/or compared to clinic-based service delivery."

Strengths and Limitations *Comment 1. These should be clearly sub-divided or narrated to show what is the stated strength and what is the stated limitation.*

- Two sub-headings have been created to separate strengths from limitations see lines 3 and 11 on page 3

Comment 2. May be helpful to state that your review focuses on the sustainability of evidence based DSDs in terms of prior evidence available on their actual impact with regard to patient outcomes?

- Suggested text incorporated in line 5 on page 3

Introduction

Comment 1. *Line 5-6, suggest: In 2015, the WHO recommended treatment initiation for all people living with HIV (PLHIV), the so-called "universal test and treat" strategy.*

- Suggested text incorporated in line 2 on page 4

Comment 2. Line 11, suggest: *DSD prioritizes patient preferences while also *aiming* to increase health system efficiency*

- Suggested text incorporated in line 7 on page 4

Comment 3. Line 36: *"when" care is provided *as* a major benefit...*

- Suggested text incorporated in line 26 on page 4

Comment 4. Page 6, Table 1 title: *Characterization of DCM Models." DCM is an undefined acronym at this point, also it stands for Differentiated Care Models and so the word "models" after that is not needed. I suggest sticking to "DSD models".*

- DCM replaced with DSD as suggested in line 2 on page 5

Comment 5. Line 46-47: *Suggest: "until a vaccine *and/or cure * is developed..."*

- Suggested text added in line 22 on page 5 in the section – Why is it important to do this review?

Comment 6. *Eligibility criteria: these could be itemized in a bullet point on a Box Table for easy reference. I suggest definition of adult should be provided by a specific age range. Also, even though no restriction on definition of stable patient, authors can state they will rely on what the reviewed studies define as stable patient and with the actual review, provide some summary of what studies defined as stable patients.*

- The text under Eligibility criteria has been itemized as bullet points in Box Table 1. The definition of adult with an age range has been specified including the suggested text in Box Table 2 (see below).

Comment 7. Page 13, line

Box 1: Eligibility criteria

Inclusions –

- Observational, experimental or quasi-experimental studies.
- Studies involving stable adult ART clients accessing HIV care in SSA.
- Studies describing or assessing HIV services delivered through models other than standard clinic-based care
- Studies which compare the performance of these other service delivery models with standard clinic-based HIV service delivery accessed by other clients. Though, lack of this comparison is not an exclusion criterion.

Exclusions –

- Reviews, editorials, protocol studies and clinical guidelines
- Studies describing or assessing interventions focussed on special population groups e.g. adolescents, children, pregnant women, men who have sex with men, commercial sex workers etc
- Studies utilising data retrospectively collected in electronic databases with little description of the actual intervention

Box 2: Definitions

Stable ART clients

The definition of a stable ART client has been fine-tuned over time, therefore, there will be no restriction in terms of definition. We will rely on what included studies define as stable patients and however with the actual review, provide a summary of definitions employed.

Adult clients

The definition of adult clients varies across studies. We will define adult as ≥18 years but will include studies if the age-range of participants is not < 15 years

5: *as countries in SSA strive to increase *the/their domestic* share of funding *for* their HIV programs...*

- Suggested text incorporated in line 3 on page 13 under Conclusion

Supplementary Files

Comment 1. *Search strategy: I assume authors will expand the search period beyond September 2018, when they actually perform the search? E.g. expand closer to Dec 31, 2019?*

- Date for final search has been extended to 30th November 2019 and reflected as such in Supplementary files

Comments 6 – References

1. *I only found one reference document published in 2018. All other refs published before that. I expect there are more papers and DSD literature/guidelines/reports published in 2018 and 2019 that are relevant to the protocol under review. Please identify these and add for more recent context.*

- We have updated the protocol with more recent papers published in the last couple of years. Extending our search date to November 2019, will ensure the inclusion of the most recent and relevant papers in the actual review

B. Reviewer #2:

I think this is an important systematic review to undertake and the authors have a concrete existing framework for sustainability to draw from.

Comment. *The only issue I have with this protocol is that at the information that the authors are looking for is likely under reported in the manuscripts, and the most likely outcome of the review is that sustainability is unclear because determinants of sustainability are rarely described. That would be a valuable observation, but more about the state of research in public health than about DSD per se. In fact, the authors should consider taking a random sample of papers to review in random order and if in the first 15 or 20 papers the information sought is vastly under-reported, the project could be stopped or it could pivot to answer a slightly different question.*

- We agree that the issue raised may arise for some studies. We have elaborated on the issue by expanding the Data extraction section in lines 3-7 on page 8 to clarify how we will address it. We agree that quantifying the extent and direction of missing information will be an important finding to share for better research reporting. To date, we have started data extraction and fortunately have found this issue less often than anticipated. We will be updating the search soon and will be able to capture the extent of the issue then but for the moment, we wanted to reassure the reviewer.

“We speculate that information regarding some constructs in the framework we will be using in our evaluation may be under-reported in the included articles. However, by reporting the extent to which this under-reporting is happening, we hope to highlight this issue for further research. Therefore, we will quantify and report the levels of missing information observed during data extraction (estimated by the number of “not described” assigned by reviewers) per construct for included studies.”

C. Other minor edits made in manuscript

Abstract

Minor edits in the introduction section to improve clarity and flow i.e.

- Deleting “so called universal test and treat” in line 4
- Deleting “beyond the traditional clinic-based care” and adding “ so called” before Differentiated service delivery to clarify what differentiated service delivery (DSD) is

Introduction

Replace calls with called in line 6 on page 4.

Include reference 17 in line 36 on page 4

Methods and Analysis

Addition of text in lines 38 and 39 explaining eligibility criteria and some definitions in Boxes 1 and 2.

Table 2: Sustainability constructs and evidence to be extracted from included papers (adapted from [29]) (Primary study outcomes)

- Inclusion of one column – “**Item**” to enable the numbering of framework constructs for easy reference

Authors contribution

Minor edits to clarify the specific role of each individual author in writing the review protocol.

VERSION 2 – REVIEW

REVIEWER	Nadia A Sam-Agudu 1. Institute of Human Virology, University of Maryland School of Medicine, USA. 2. Institute of Human Virology Nigeria, Abuja, Nigeria
REVIEW RETURNED	08-Dec-2019
GENERAL COMMENTS	Thank you for your detailed revisions and response. My comments have been satisfactorily attended to. The protocol reads much better now, and methods much more rigorous. I appreciate the addition of

	an assessment of missing data to report as part of the protocol. I have no further major feedback to provide.
--	---